# Preparation and Characterization of Erythrocyte Membrane-Camouflaged Berberine Hydrochloride-Loaded Gelatin Nanoparticles

**DOI:** 10.3390/pharmaceutics11020093

**Published:** 2019-02-22

**Authors:** Jing Su, Ran Zhang, Yumei Lian, Zul Kamal, Zhongyao Cheng, Yujiao Qiu, Mingfeng Qiu

**Affiliations:** 1School of Pharmacy, Shanghai Jiao Tong University, Shanghai 200240, China; jingsu@sjtu.edu.cn (J.S.); zhangran0309@sjtu.edu.cn (R.Z.); lym-0517@sjtu.edu.cn (Y.L.); zulkamalsjtui@sjtu.edu.cn (Z.K.); zhongyaocheng@sjtu.edu.cn (Z.C.); 2Department of Pharmacy, Shaheed Benazir Bhutto University, Sheringal Dir (Upper), Khyber Pakhtunkhwa 18000, Pakistan; 3Solebury School, New Hope, PA 18938, USA; louisaqiu0712@gmail.com

**Keywords:** erythrocyte membrane, berberine hydrochloride, gelatin, nanoparticles

## Abstract

The discovery of a new pharmacological application of berberine hydrochloride (BH) made it more clinically valuable. However, the further development of BH was hampered by its short half-life and side effects after intravenous injection. To overcome these problems, a novel BH delivery system was developed using natural red blood cell membrane-camouflaged BH-loaded gelatin nanoparticles (RBGPs) to reduce the toxicity associated with injections and achieve sustained release. The size of the RBGPs was 260.3 ± 4.1 nm, with an obvious core–shell structure, and the membrane proteins of the RBGPs were mostly retained. The RBGP system showed significant immune-evading capabilities and little cytotoxicity to human embryonic kidney (HEK) 293T cells and LO2 cells. Finally, RBGPs improved the sustained releasing effect of BH significantly. When the cumulative release time reached 120 h, the cumulative release rate of RBGPs was 78.42%. In brief, RBGPs hold the potential to achieve long circulation and sustained-release of BH, avoid side effects caused by high plasma concentration in common injection formulations, and broaden the clinical applications of BH.

## 1. Introduction

Berberine hydrochloride (BH, C_20_H_18_ClNO_4_, molecular weight (*M*_W_) = 371.82) is a well-known active isoquinoline alkaloid widely used in traditional Chinese medicine (TCM). It is commercially used for diarrhea caused by bacteria because of its antimicrobial, antimotility, and antisecretory properties. As an over-the-counter (OTC) antidiarrheal drug, it has few oral adverse reactions, and is the first choice for bacillary dysentery in the clinic [1]. Recent studies have demonstrated that BH also possesses other pharmacologic properties such as anti-malaria [2], anti-hypertension [3], anti-hyperlipidemia [4], anti-hyperglycemia [5], anti-arrhythmia [3], anti-tumor [6], anti-inflammatory [7], and neuro protective effects [8]. It has good clinical effects and can be taken orally, but its bioavailability is low, with a short half-life [9]. In addition, it can quickly distribute into organs and tissues due to its intramuscular and intravenous administration, and cause various side effects such as vasodilatation, hypotension, and cardiac inhibition. [9]. Since its traditional oral dosage forms and injection preparations have limited its therapeutic applications, there is an intense need to develop new formulations to overcome the above problems.

Supramolecular materials based on nano-carriers have significant prospective medical applications which can improve the physical and chemical properties of drugs, achieve long circulation sustained-release effects as well as targeted therapy, and increase the bioavailability of drugs [10]. The reported nano-dosage forms of BH include polymer-based nanoparticles [11], solid liposome nanoparticles [12], and gold nanoparticles [13]. Polymer–lipid hybrid nanoparticles (PEG–lipid–PLGA NPs) loaded with berberine (BBR) phospholipid complex were prepared using a solvent evaporation method to enhance oral efficiency. The BBR–soybean phosphatidylcholine complex (BBR–SPC) could enhance the liposolubility of BBR and improve the affinity with the biodegradable polymer to increase the drug-loading capacity and controlled/sustained releasing effects [11]. BBR-loaded solid lipid nanoparticles (Ber-SLN) composed of Cremophor EL and glyceryl behenate were prepared using a high-pressure homogenization technique. It is a promising approach for treating type 2 diabetes [12]. However, the reported nano-dosage forms of BH still have low oral bioavailability and potential side effects after intravenous injections [9]. In particular, they require long and frequent administration for the treatment of cardiovascular disease and cancer.

Gelatin is a kind of natural biodegradable polymer material. Its raw material is easy to obtain and its preparation process is simple [14]. Gelatin nanoparticles loaded with moxifloxacin were prepared using a modified two-step desolvation method. This method could be used for effective ocular delivery and controlled release of drugs in the corneal eye layer [15]. Gelatin (Type A, derived from acid cured tissue) was modified by crosslinking with genipin, and nanoparticles of crosslinked gelatin were prepared using single water in oil (W/O) emulsion technique. The biocompatible gelatin nanoparticles were capable of releasing the cytarabine in a controllable way by regulating the extent of swelling of nanoparticles [16]. As a drug carrier, gelatin has an obvious sustained release effect and certain tissue-targeting characteristics, which can increase the therapeutic effects while reducing associated side effects [17]. 

In the past few years, coating nanoparticles with cell membranes for treatment has attracted attention. Different cell membranes, such as T-cell-membrane [18], plasma membranes of gastric epithelial cells [19], and blood cell membranes [20], have been reported for the development of novel nanotherapeutics. Normally, human erythrocytes have a lifespan of 100–120 days (almost three times the mouse erythrocyte lifetime) and travel ~250 km throughout the cardiovascular system, functioning as natural carriers for oxygen [21]. The red blood cell membrane (RBCM) can be relatively easily extracted by the hypotonic hemolysis method [22], inheriting all the superiorities of erythrocytes, such as good biocompatibility and biodegradability and a long circulation life span. Celecoxib-encapsulated erythrocyte membranes (CB-RBCMs) were reported to have a sustained release of CB over a period of 72 h in vitro and exhibit high brain biodistribution efficiency following intranasal administration, providing a rational design to treat Alzheimer’s disease (AD) by promoting the self-repairing capacity of the brain [23]. RBC-mimetic NPs composed of a paclitaxel (PTX)-loaded polymeric core and a hydrophilic RBC vesicle shell were also reported. The results revealed the importance of both long circulation and tumor penetration of nanosized drugs for efficient cancer therapy, which provide a new insight for nanosized drug delivery systems (NDDSs) designed [24]. Collectively, certain sized drug-loading nanoparticles can be encapsulated into the RBCM to achieve sustained release, prolong half-life, reduce toxicity, and improve bioavailability [25].

In order to avoid the deficiency of the existing nano-dosage forms of BH, red blood cell membrane-camouflaged berberine hydrochloride-loaded gelatin nanoparticles (RBGPs) were prepared to achieve sustained releasing effects and reduce the toxicity associated with injections of BH in this paper (Figure 1). The RBGPs had a uniform spherical nano-sized structure and showed a core–shell structure with the RBCM coating. The RBGPs also showed significant immune-evading capabilities and little cytotoxicity to HEK 293T cells and LO2 cells. Moreover, they slowed down the sustained releasing property. The above results demonstrated that RBGPs have the potential to be a promising delivery system to achieve prolonged circulation and sustained release of BH.

## 2. Materials and Methods 

### 2.1. Materials

BH (HPLC, ≥99%) was purchased from Meilun Biologics (Dalian, China). Gelatin (~250 g Bloom) and sodium metabisulfite (HPLC, ≥99%) were obtained from Macklin (Shanghai, China). Glutaraldehyde solution (50 wt. % in H_2_O) and ethylenediaminetetraacetic acid (EDTA, HPLC, ≥99.5%) were purchased from Innochem (Beijing, China). Phosphate-buffered saline (PBS, 500 mL), 2-(4-amidinophenyl)-6-indolecarbamidine dihydrochloride (DAPI, 5 mg/mL), and the sodium dodecyl sulfate polyacrylamide gel electrophoresis (SDS-PAGE) gel quick preparation kit were obtained from Beyotime Biotechnology (Shanghai, China). Fluorescein isothiocyanate (FITC, 10 mg), Cy5-NHS (5 mg), and hydrophobic 1,1′-dioctadecyl-3,3,3′,3′-tetramethylindocarbocyanine perchlorate (Dil, 10 mg) were purchased from Tianjin Biolite Biotech Co., Ltd. (Tianjin, China). A dialysis bag with a 3000 molecular weight cut off was purchased from Sigma-Aldrich Chemicals Private Ltd (Bangalore, India). All other chemicals were analytical grade.

### 2.2. Cell Culture

The human embryo kidney cells (HEK 293T) were purchased from the Institute of Cell Biology, Chinese Academy of Sciences (Shanghai, China) and cultured in dulbecco’s modified eagle medium (DMEM) supplemented with 10% fetal bovine serum (FBS). Cells were maintained at 37 °C in a humidified 5% CO_2_ incubator. The human hepatocytes (cell line LO2), and mouse peritoneal macrophages (RAW 294.7) were purchased from the Institute of Cell Biology, Chinese Academy of Sciences, and cultured in 1640 medium supplemented with 10% FBS. Cells were maintained at 37 °C in a humidified 5% CO_2_ incubator.

### 2.3. Preparation of BH-Loaded Gelatin Nanoparticles (BH-GNPs, BGPs)

The BGPs were prepared using the desolvation method according to the literature with some optimization [17]. Gelatin aqueous solution (0.5% *w*/*v*, 20 mL) was prepared by stirring the solid gelatin in ultrapure water of 50 °C at 1000 rpm to obtain a clear solution. The pH was set at 6.0 by the addition of NaOH (0.2 M). The solution was subsequently filtered after being cooled down to room temperature. Then, 10 mg of BH were added into the resulting gelatin solution and ultrasonicated to dissolve. Then, 20 mL of acetone were added to obtain a permanent faint turbidity. Finally, glutaraldehyde aqueous solution (10% *v*/*v*, 500 μL) was added to harden the particles and then stirred for 2 h under 50 °C at 1000 rpm. The crosslinking was stopped by the addition of aqueous sodium metabisulfite solution (1.6% *w*/*v*, 8 mL). The nanoparticle solution was dialyzed overnight to remove the unloaded BH to obtain BGPs. The Cy5-labeled gelatin nanoparticles (GPs) were prepared by adding 200 μL Cy5-NHS solution (1 mg/mL) into 20 mL of gelatin solution. In order to remove the excess Cy5, the solution was washed with PBS by centrifuging (13,200 rpm, 10 min, 4 °C); this was repeated three times.

### 2.4. RBCM Preparation 

The RBCM was prepared using hypotonic hemolysis method. The whole blood was collected from Sprague-Dawley rat (SD rat) by apical puncture method with 0.25 mg heparin sodium per mL of blood for anticoagulation. The blood was centrifuged at 3800 rpm and 4 °C for 10 min to remove the plasma and the buffer coat. The resulting RBCs were washed three times with 1× PBS. Then, 2 mM of EDTA were added for hemolysis. The released hemoglobin was removed by centrifuging at 13,200 rpm for 10 min at 4 °C, and the RBCM was collected and washed with 1× PBS four times. The resulting RBCM was stored in a refrigerator (−80 °C). All procedures performed where animal were involved were in accordance with the Guidelines for Care and Use of Laboratory Animals of Shanghai Jiao Tong University and approved by the Animal Ethics Committee (Number: A2018021, 26 March 2018).

### 2.5. Preparation, Characterization, and Stability Test of RBGPs

The collected RBCM was ultrasonicated at 100 W for 3 min. Then the resulting RBCM-derived vesicles (RVs) were extruded through 400 nm and 200 nm polycarbonate membranes (Whatman, Shanghai Nalujie Biotechnology Co., Ltd., Shanghai, China) using a mini-extruder (Avanti Polar Lipids Inc., Shanghai Nalujie Biotechnology Co., Ltd., Shanghai, China). RBGPs were prepared by mixing and extruding RVs and BGPs through 200-nm polycarbonate porous membranes repeatedly at the surface area ratio of 1:2 [25]. Newly prepared RBGPs were stored at 4 °C. The size and polydispersion of BGPs were analyzed by dynamic light scattering with particle size apparatus (Malvern Zetasizer, Malvern, UK) for 30 days and RBGPs for 7 days.

### 2.6. Determination of Loading Capacity and Encapsulation Efficiency of BH in BGPs by HPLC

The content of BH in BGPs was determined by HPLC. BGPs were dialyzed (cellulose membrane, *M*_W_ cutoff 3000 Da) overnight and the unloaded BH solution was collected. Then 20 μL dialysis solutions were automatically injected into HPLC system. Separations were performed on a ZORBAX SB-C18 column (4.6 mm × 250 mm, 5 μm), and effluents were monitored at 345 nm. The mobile phase was acetonitrile and 0.05 mol/L KH_2_PO_4_ solution containing 0.4% (*w*/*v*) sodium dodecyl sulfate (adjusting pH value to 4 by adding phosphoric acid) (50:50, *v*/*v*). A flow rate of 1.0 mL per minute was used for both pharmaceutical ingredients. Drug loading capacity (%) and encapsulation efficiency (%) of BH were calculated according to the following equations: Encapsulation Efficiency (%) = *W*_2_ − *W*_1_/*W*_2_ × 100%
Drug Loading Capacity (%) = *W*_2_ − *W*_1_/*W*_3_ × 100%
where *W*_1_ is the mass of free BH, *W*_2_ is the mass of initial added BH, and *W*_3_ is the total mass of the gathered BGPs.

### 2.7. Morphology Test of RBGPs

The morphologies of RBGPs were observed using transmission electron microscope (TEM). The samples were placed on a carbon-coated copper grid to leave a thin film and then negatively stained with 1% phosphotungstic acid. The sample grid was dried thoroughly at room temperature and observed using a Biology Transmission Electron Microscope (Tecnai G2 SpiritBiotwin/*Tecnai G2 spirit Biotwin, Instrumental Analysis Center of SJTU, Shanghai, China) under appropriate magnification at 120 kV.

### 2.8. Membrane Protein Retention of RBCM with Sodium Dodecyl Sulfate Polyacrylamide (SDS-PAGE) Gel Electrophoresis 

Prepare the separation and concentrated gels with a concentration of 8% and 5% respectively. The protein concentration of each sample was determined by BCA protein quantification kit and protein concentrations of RBCM, GPs, and RBCM-coated GPs (RBCM-GPs, RGPs) were adjusted to same concentrations. Then 100 μL of the above groups were mixed with 25 μL loading buffer, respectively. The mixtures were boiled in water bath at 95 °C for 5 min to denature the proteins. Then, the prepared gel plate was loaded with 15 μL of each group and 2 μL of protein marker were added into another separate loading slot. After the electrophoresis, the gel pieces were stained with Coomassie Brilliant Blue at room temperature for 30 min in the dark and decolorized overnight on decolorizing shaker. The gel was recorded by photographing under the Tanon2500 Gel Imaging Analysis System (Shanghai Tianneng Technology Co., Ltd., China). 

### 2.9. Macrophage Uptake Study

The immune-evading capability of RBGPs was examined by anti-phagocytosis against RAW 264.7 macrophage cells. RAW 264.7 cells were cultured in 1640 medium with 10% FBS and then seeded in a 24-well plate with a density of 10^5^ cells per well. After incubation for 24 h, the Cy5-labeled RGPs and GPs (excess Cy5 was washed three times with PBS) were added into the culture medium. The nanoparticles were incubated with RAW 264.7 cells for another 2 h and then washed with PBS to examine the macrophage uptake of Cy5-labeled nanoparticles. The uptake of nanoparticles by RAW 264.7 cells was imaged with a confocal laser scanning microscope.

### 2.10. Cytotoxicity Assay of RBGPs

Human embryonic kidney 293T cells were cultured in DMEM medium (50 mL of DMEM mixed with 10% FBS and 1% penicillin and streptomycin solution) and human hepatocytes LO2 cells were cultured in 1640 medium (50 mL of 1640 medium mixed with 10% FBS and 1% penicillin and streptomycin solution). HEK 293T cells and LO2 cells were incubated with different concentrations of RBGPs (30, 60, 120, 240, 300, 480, 600, and 750 μg/mL, administration groups(AG)) for 24 h in a 96-well plate and then 10 μL Cell Counting Kit-8 (CCK-8) solution was added after replacement of fresh DMEM medium (90 μL), and the samples were incubated for another 4 h. DMEM medium cells and CCK-9 were taken as a control group (CG), where DMEM medium and CCK-8 were set as blank group (BG). The absorbance at 450 nm with a reference wavelength of 650 nm was determined with a multifunctional microplate reader.
Cell Survival Rate (%) = (OD_AG_ − OD_BG_)/(OD_CG_ − OD_BG_) × 100%

### 2.11. Hemolysis Test of RBGPs

The RBCs were collected from the blood of SD rat. A saline solution was added to prepare 2% (v/v) of red blood cell suspension. The red blood cell suspension, physiological saline, distilled water and RBGPs were added to seven tubes according to Table 1. The results were recorded after 3 h and 24 h of incubation at 37 °C, respectively. The hemolysis rate of supernatant was measured by ultraviolet spectrophotometry at 540 nm wavelength.

### 2.12. In Vitro Release Test 

The release profiles of BGPs were performed in saline phosphate buffer (PBS, pH 7.4) using the dialysis bag method. The dialysis membranes (cellulose membrane, *M*_W_ cut off 3000 Da) were soaked overnight in the dissolution media in advance to ensure completely the membrane wetness. Similarly 2 mL of BGP solution were placed into the dialysis bag tied and fixed by clamps on both ends. The dialysis bag was submerged into a beaker containing 20 mL of PBS at 37 ± 0.5 °C and stirred magnetically at 100 rpm. Samples were extracted at preset intervals with immediate replacement of equal volumes of fresh PBS to maintain sink level. The samples were filtered with 0.22-mm syringe filters and BH content was determined by HPLC. A similar study was also conducted with RBGPs and pure BH solution. All the tests were carried out in triplicate.

### 2.13. Statistical and Data Analyses

Data expression was shown as mean ± SD. Significant differences between RBGPs and BGPs were analyzed by a Tukey Kramer multiple comparison test, using GraphPad Prism Software, v.6.01 (GraphPad Software, Inc., San Diego, CA, USA. Available: http://www.graphpad.com. Accessed 2014.). Results with *p* < 0.05 were considered to indicate a significant difference and very significant difference was considered at *p* < 0.01.

## 3. Results and Discussion

### 3.1. Characterization and Stability Test of BGPs and RBGPs

BGPs were prepared using a reported desolvation method with some optimizations [17]. RBGPs were prepared by mixing and repeatedly extruding RBCM and BGPs through 200 nm polycarbonate porous membranes at the surface area ratio of 1:2. Dynamic light scattering (DLS, Figure 2) results indicated that the hydrodynamic diameters of BGPs increased from 243.6 ± 3.7 nm to 260.3 ± 4.1 nm upon coating with RBCM. The thickness of the coating membrane was approximately 8 nm, which implies a ~8 nm thick lipid shell formed, considered as the membrane thickness of RBCs. Moreover, the stability of the BGPs and RBGPs in PBS was monitored by DLS. For BGPs, after incubating for 1 day, the particle size of BGPs was 240.0 ± 4.1 nm, and after a 30-day time interval, no significant particle size difference of BGPs could be observed, and the particle size was 250.8 ± 2.8 nm (Figure 3A). The PDIs of BGPs were checked at different time intervals; all PDIs were below 0.2, which suggested that BGPs were well dispersed and had a good quality after 30 days (Figure 3B). The stability assay showed that BGPs were stable for almost one month and were a suitable core for the next RBCM coating process. Then the stability of RBGPs was investigated. After 1 days, the particle size of RBGPs was 256.1 ± 2.9 nm, and no significant particle size difference was observed after 7 days (264.3 ± 1.5 nm), as shown in Figure 3C. The results showed RBGPs were stable for 7 days. We also monitored the PDI of RBGPs; all the RBGPs were well dispersed and had good quality even after 7 days of incubation in PBS (Figure 3D). The results demonstrated RBGPs were suitable for the sustained release of BH.

To visualize the coating of the RBCM onto BGPs, transmission electron microscopy (TEM) was used to observe the morphology of RBGPs. As shown in Figure 4, all the RBGPs were uniformly distributed and shaped as spherical with edges, which demonstrated that a spherical core–shell structure was formed. The thickness of the coating membrane was approximately 7 nm, which was consistent with the results of DLS. Free BH was removed by dialysis overnight and the resulting BGPs were coated with RBCM. The encapsulation efficiency (EE) and loading capacity (LC) were determined by HPLC. The optimal formulation of RBGPs with an EE of 65.78% and DLC of 6.36% was obtained by mixing BH and gelatin at the ratio of 1:10 (*w*/*w*) for use in the following experiments.

### 3.2. Membrane Protein Retention of RBCM Confirmed by SDS-PAGE

As shown in Figure 5, protein bands of natural RBCM group were almost same as that of the RGPs group, which means the membrane proteins of the RBCM were retained after the sonication treatment. This finding suggests that the associated membrane proteins were also transferred to the nanoparticle surface when the translocation of the bilayer cellular membranes occurred. We can expect that biological characteristics and function of the RBCMs could also be maintained, which could help the NPs to escape the immune system.

### 3.3. Macrophage Uptake Test

SDS-PAGE gel electrophoresis test was performed to confirm that the function of RBCM was not influenced by the preparation process of the RBGP system. However, more illustrations were still needed to confirm that the bioactivity of RBCM was retained on RBGPs after the preparation process. The escaping nature of RBC from the immune system is one of the most important bio-properties of its surface membrane [18]. A macrophage uptake experiment was designed to verify that the RBGP system is able to escape from the immune system. RAW 264.7 macrophage cells were selected as the model immune cells. Hydrophilic fluorescent Cy5-NHS probes were loaded into polymeric core to form Cy5-GPs. As shown in Figure 6, red fluorescence (labeled out with white arrows) appeared in RAW 264.7 cells in the BGP group, while no fluorescence appeared in RBGPs group. The merged pictures showed that the Cy5-GPs were internalized by RAW 264.7 macrophage cells, while RBCM-Cy5-GPs were barely internalized. The result indicated that RBGPs can avoid the recognition by macrophage cells, which implies the RBGPs hold the potential to fulfill the prolonged circulation function in vivo. 

### 3.4. Cytotoxicity Assay of RBGPs

An in vitro cytotoxicity assay was performed as a preliminary safety assessment of RBGPs. Human embryonic kidney (HEK 293T) cells and human hepatocytes (LO2 cells) were used to predict the liver and kidney toxicity of RBGPs in vivo. HEK 293T and LO2 cells were incubated with different concentrations of RBGPs (30, 60, 120, 240, 300, 480, 600, and 750 μg/mL) for 24 h. The RBGPs showed high cell viability (90.58% in HEK 293T cells, 90.28% in LO2 cells) even at the highest concentration (750 μg/mL) (Figure 7). This demonstrated that RBGPs had little cytotoxicity to HEK 293T cells and LO2 cells, which implied that the erythrocyte membrane-derived drug delivery system had good biocompatibility.

### 3.5. Hemolysis Test of RBGPs

Since RBGPs were administered by intravenous injection, it was essential to evaluate whether RBGPs could cause hemolysis or aggregation prior to injection. A hemolysis test was designed to study the safety of the system. The results show that the solution of positive control (distilled water) group is totally red and clear without cell residue at the bottom (Figure 8A), suggesting that hemolysis occurs in RBCs. For the RBGP group, with the concentration increasing from low to high, all the red blood cells sank, and the supernatants were colorless and clear (Figure 8B). Further observation showed that there was no erythrocyte aggregation under an inverted microscope. The percentage of hemolysis measured by ultraviolet spectrophotometer also showed that for each group it was less than 0.1%, and when the effective dose was in the intermediate concentration range, the hemolysis rate was less than 0.05%. This could be considered as no hemolysis. The hemolysis test indicates RBGPs will not cause hemolysis or aggregation of RBCs, and are suitable for intravenous injection. 

### 3.6. Release Profiles of BH from RBGPs and BGPs

Release profiles of RBGPs and BGPs were performed using dialysis method. The RBGPs and BGPs were incubated in PBS for 120 h. As shown in Figure 9 at time intervals of 0.5 h, 65.15%, 24.60%, and 38.56% of BH were released from the free BH group, the BGP group, and the RBGP group, respectively. The highest burst release rate was observed in free BH group. In the beginning period (0–2 h), all the BH was released in the free BH group and decreased drug release efficiency was observed in the BGP group compared to RBGP group. Little BH might be leaked out from the BGP group under sonication treatment. In addition, a slightly decreased drug release efficiency was found in the RBGP group compared to BGs group from 4 h to 120 h. In total, 65.50% of BH was released in the BGP group within 4 h, while less than 60% of BH was released from the RBGP group. Finally, the cumulative BH release from BGP group and RBGP group reached 82.46% and 78.42% within 120 h, respectively. All these results implied that the release of BH was slowed in the BGP group, and the release of BH from RBGP would be further lowered by the physical barrier around the nanoparticles formed by the RBCM.

## 4. Conclusions

In summary, a new RBCM-derived drug delivery system is reported to slow down the release rate of BH. Within 30 days, the BH-loaded gelatin nanoparticles (BH-GNPs, BGPs) were stable and showed a uniform spherical structure, and the average encapsulation efficiency and average loading percentages of the BGPs were 65.78% and 6.36%, respectively. The surfaces of the gelatin nanoparticles (G-NPs, GPs) were then wrapped with monolayers of RBCM by extrusion. For almost 7 days, the RBCM-coated GPs (RBCM-GPs, RGPs) were able to maintain their basic structure and showed an apparent core–shell structure. The release of BH was slowed down in both RBGPs and BGPs, and its sustained release effect was further strengthened by the physical barrier formed by the RBCM outside the BGPs. The cumulative release rate of BH from the RBCM-BH-GNPs at 120 h was 78.42%. In general, the biological characteristics of the natural RBCM were maintained in the RBGPs which could avoid recognition by macrophage cells and had no obvious toxicity to HEK 293T cells and LO2 cells in vitro, and could not cause hemolysis or aggregation of red blood cells. This demonstrates the potential of the developed RGBP system to overcome the weaknesses of BH such as its rapid metabolic decomposition in vivo and lack of sustained release effect by general injection.

## Figures and Tables

**Figure 1 pharmaceutics-11-00093-f001:**
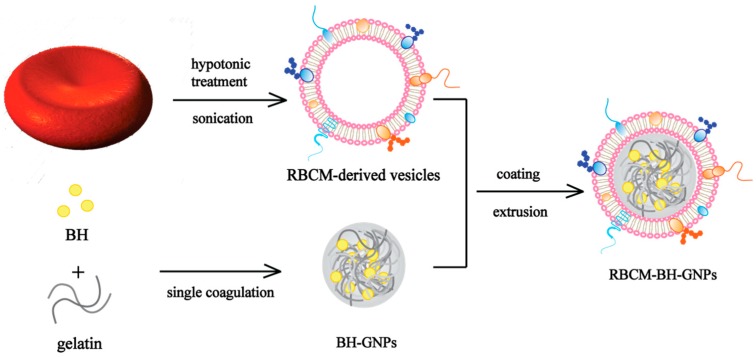
Schematics of preparation process of the erythrocyte membrane-camouflaged berberine hydrochloride-loaded gelatin nanoparticle (RBGP) system. RBCM: red blood cell membrane; BH: berberine hydrochloride; BH-GNPs: BH-loaded gelatin nanoparticles.

**Figure 2 pharmaceutics-11-00093-f002:**
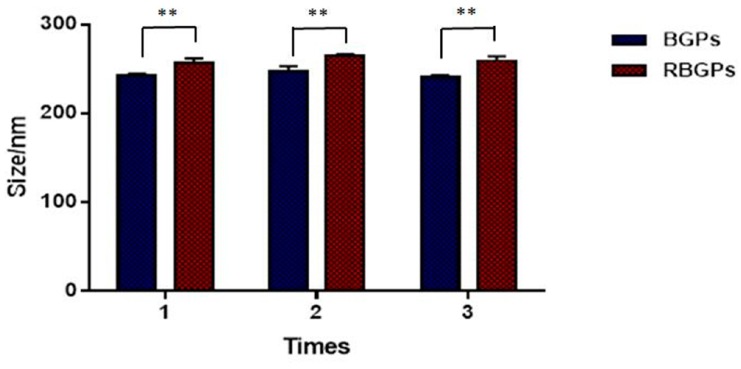
Size comparison between the BGPs and the RBGPs, repeated three times. Data are represented as mean ± SD, and “**” corresponds to *p* < 0.01.

**Figure 3 pharmaceutics-11-00093-f003:**
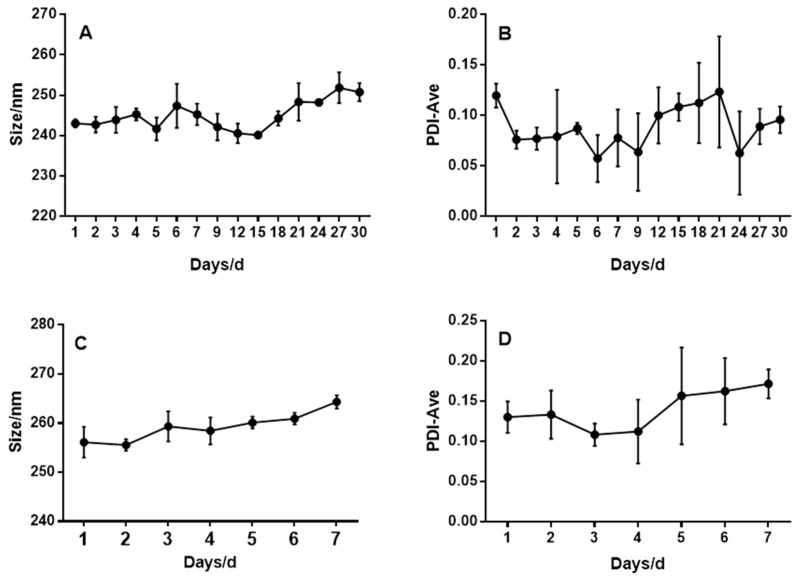
Stability test of BGPs and RBGPs. Size (**A**) and PDI (**B**) stability of BGPs over a span of 30 days. Size (**C**) and PDI (**D**) stability of RBGPs over a span of 7 days. All values are expressed as mean SD (*n* = 3), and the test was repeated independently in triplicate.

**Figure 4 pharmaceutics-11-00093-f004:**
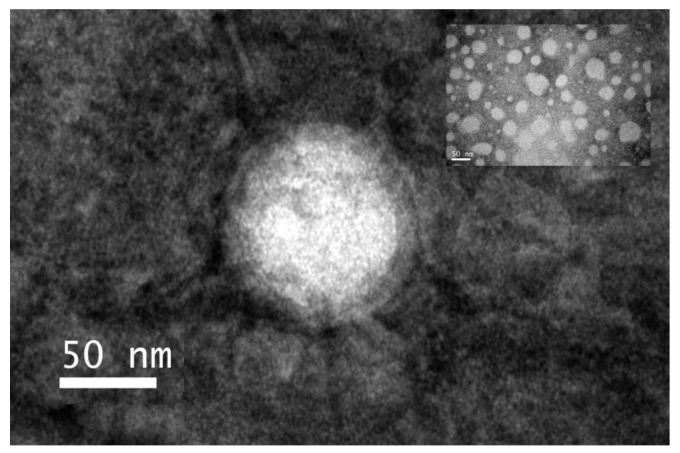
The distribution and the core–membrane structure image of RBGPs observed by TEM.

**Figure 5 pharmaceutics-11-00093-f005:**
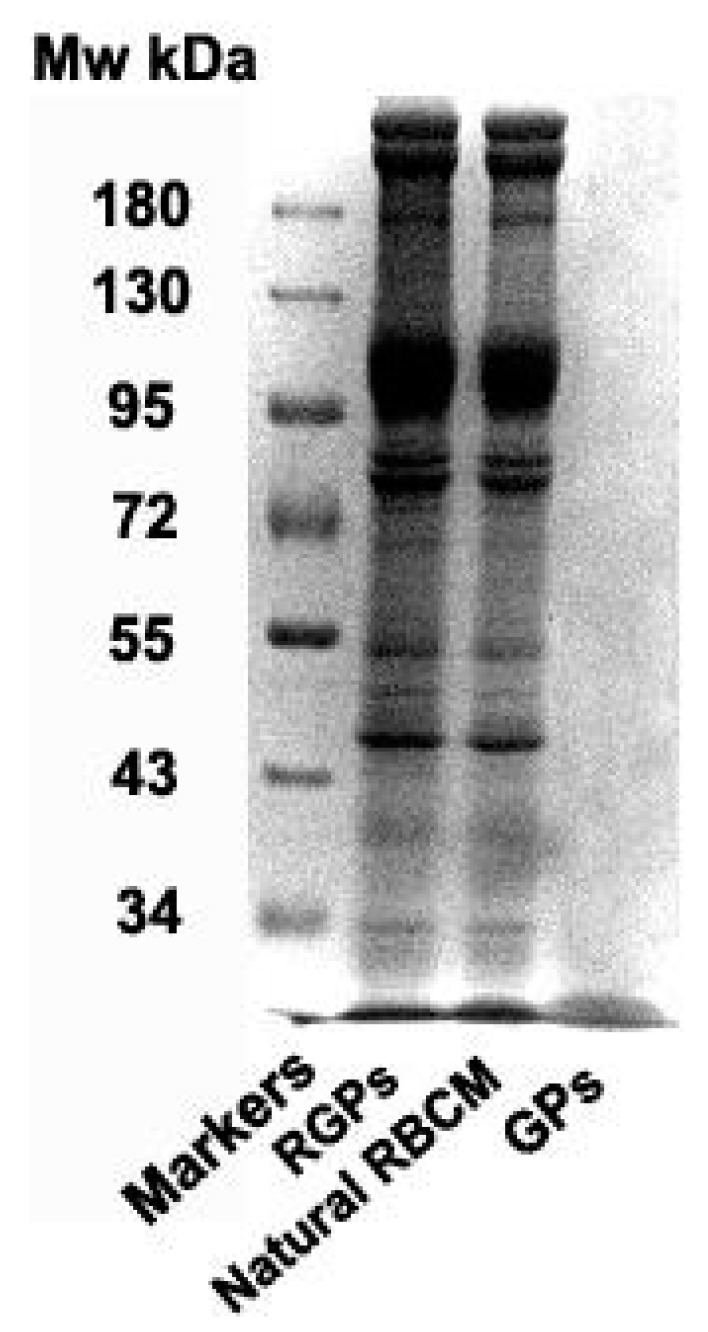
Protein detection in RGPs, natural RBCM, and GPs by SDS-PAGE stained with Coomassie Brilliant Blue and photographed under the Tanon 2500 Gel Imaging Analysis System.

**Figure 6 pharmaceutics-11-00093-f006:**
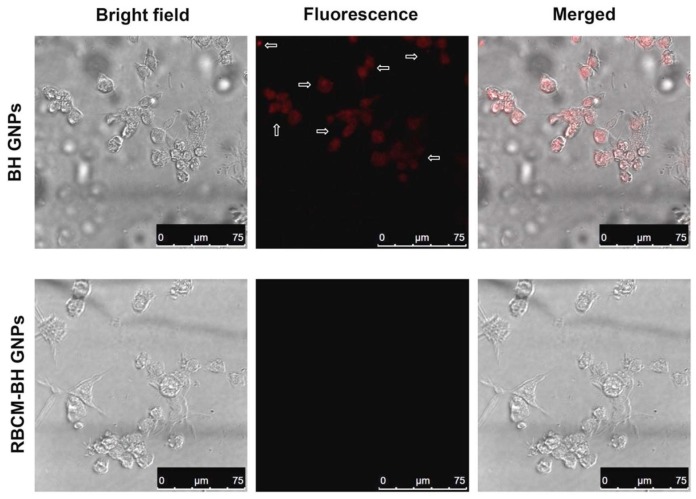
Fluorescence images of RAW 264.7 macrophage cells treated by RBGPs and BGPs, respectively. The GNPs were labeled with Cy5 (red).

**Figure 7 pharmaceutics-11-00093-f007:**
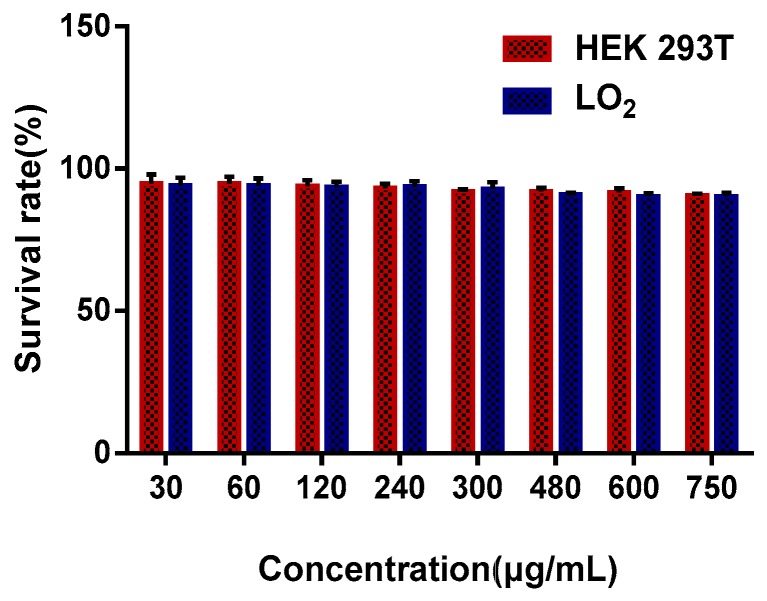
Cell viability of HEK 293T cells and LO2 cells upon treatment with a series of concentrations of RBGPs (3, 60, 120, 240, 300, 480, 600, 750 μg/mL).

**Figure 8 pharmaceutics-11-00093-f008:**
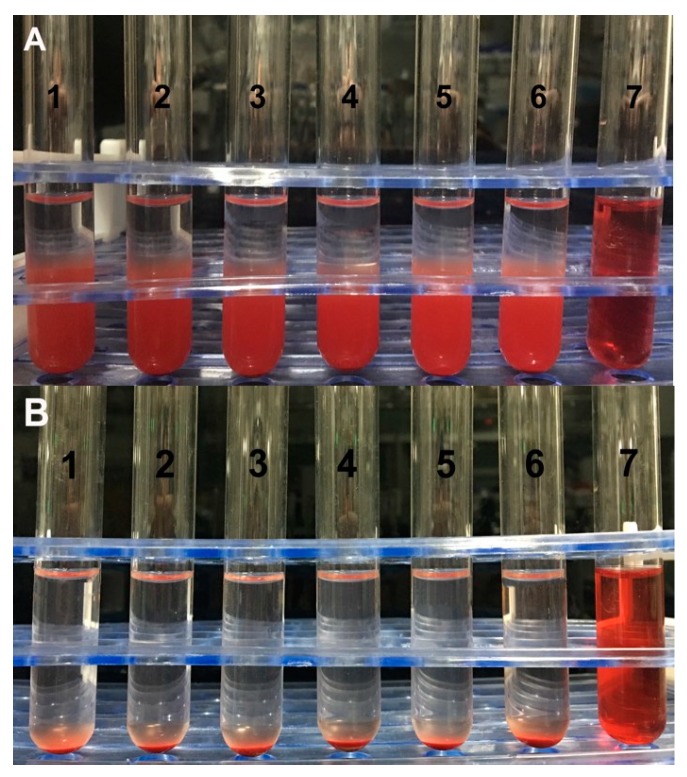
Photographs of the anti-hemolysis of RBGPs with different concentrations. (**A**) represents 3 h of incubation at 37 °C, (**B**) represents 24 h of incubation. The concentrations of RBGPs were 2% (1), 4% (2), 6% (3), 8% (4), and 10% (5). Tube 6 was from the physiological saline group and tube 7 was from the distilled water group.

**Figure 9 pharmaceutics-11-00093-f009:**
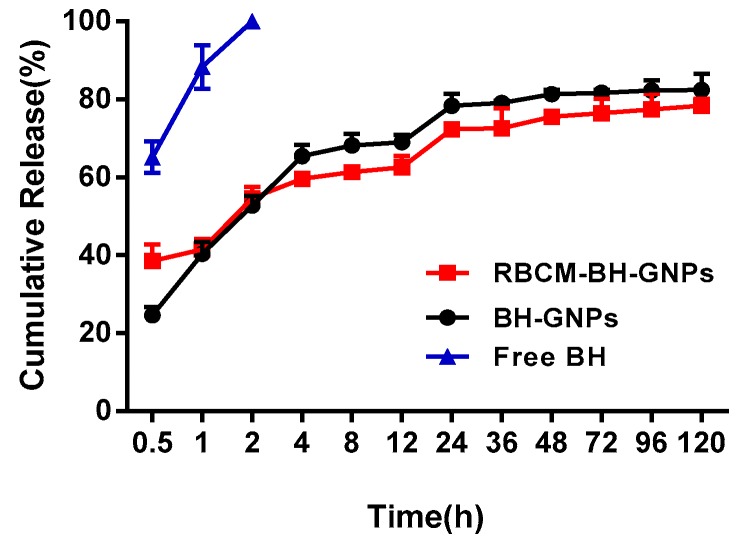
Release profiles of RBGPs and BGPs at 37 °C for 120 h; experiments were repeated independently at least three times.

**Table 1 pharmaceutics-11-00093-t001:** Groups of hemolysis test of RBGPs.

Group Number	1	2	3	4	5	6	7
2% RBC suspension (mL)	2.5	2.5	2.5	2.5	2.5	2.5	2.5
Physiological saline (mL)	2.4	2.3	2.2	2.1	2.0	2.5	0
Distilled water (mL)	0	0	0	0	0	0	2.5
Samples (mL)	0.1	0.2	0.3	0.4	0.5	0	0

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
