# Peer review of "Preparation and Characterization of Erythrocyte Membrane-Camouflaged Berberine Hydrochloride-Loaded Gelatin Nanoparticles"

_pharmaceutics, 2019, doi:10.3390/pharmaceutics11020093_

Round 1

Reviewer 1 Report

The authors prepared a very informative manuscript on this topic. The text is sound, and the manuscript is well written.

 I recommend this manuscript for publication with one minor correction. It is necessary to supplement the information presented in the figure with data on the percentage of hemolysis.

Author Response

Dear Reviewer,

Thank you very much for your letter and the comments from the referees about our paper submitted to pharmaceutics (Manuscript ID: pharmaceutics-408160)! We have checked the manuscript and revised it according to the comments. We submit here the revised manuscript as well as a list of changes.

If you have any question about this paper, please don’t hesitate to let me know.

Sincerely yours,

Mingfeng Qiu

Point 1:The authors prepared a very informative manuscript on this topic. The text is sound, and the manuscript is well written.I recommend this manuscript for publication with one minor correction. It is necessary to supplement the information presented in the figure with data on the percentage of hemolysis.

Response 1: The percentage of hemolysis was measured by ultraviolet spectrophotometer. Each group was less than 0.1%, and the effective dose was in the intermediate concentration range, the hemolysis rate was less than 0.05%. It could be considered as no hemolysis.

Reviewer 2 Report

The authors of the manuscript titled Preparation and characterization of erythrocyte membrane-camouflaged berberine hydrochloride-loaded gelatin nanoparticles have used a good drug delivery strategy utilizing gelatin nanoparticles for sustained release of the drug over time and a coat of RBC membranes to prevent recognition by macrophages. Hence such an approach could possibly increase circulation time of berberine and hence its therapeutic efficacy by maintaining a sustained release profile of the drug over a period of a week. While this approach seems feasible and exciting there are major concerns about this work that will need to be addressed before this can be accepted for publication.

Major comments:

1.       The authors claim that an obvious core-shell structure was observed by TEM of the particles. However, from the current images this is not visible. Furthermore, the size of the particles seems smaller than 200 nm in the TEM images, barely spherical and very polydisperse. If there are images that better reflect the population, please use those.

2.       In the preparation of Cy5 labeled gelatin, how is the unbound Cy5 removed?

3.       In Figure 2, what is the X- axis? It might be a better representation to show averages and sems of size and PDI in a table. Secondly, is there a statistically significant increase in size on coating membranes? Otherwise the conclusions that size increases on coating with the RBC membranes by 16 nm does not hold true- It might just be a variability of the measurement.

4.       In the stability testing, the authors test up to day 7 for the RBGPs- What happens after day 7?

5.       Method 2.8 is not clear. The SDS PAGE protocol has not been described nor the method of detection of the protein bands. Since this is a protein-denaturing protocol, the only information it yields is about the relative content of the proteins in the preparations, not about the composition or the distribution, maintenance of structure etc. Hence we cannot predict based on this that the biological characteristics and function of RBC-membranes is maintained in the nanoparticles, as the authors claim in several instances.

6.       In the in vitro cellular uptake studies, why would you expect these particles to be taken up by hek cells- there are no targeting molecules or internalization cues on the surface of the particles. In fact the RBGPs would appear as RBCs to the cell and RBCs do not normally internalize into cells. Why would they bind to the surface of HEK cells? In fact, based on the fluorescence microcopy images, they do not appear to internalize (Fig 6). Are there controls of GPs alone, RBCMs alone as well to really understand what is going on? Furthermore, in the DAPI images, there is non-nucleus associated blue staining on the membranes of cells- Why is this so? Overall it is not clear how this experiment would fit in with the hypothesis.

7.       All legends need more description. Figures and legend should be stand-alone data. However in this case, only a title is visible. Please include a brief description of the method (time of incubations, type of readout) statistics for the graphs etc. following the title.

8.       In figure 7, the images of the fluorescent channel are very dim- It is not clear if there is any fluorescent signal. Are there some brighter or high magnification images of the same? It should tell not only about association of the particles with the cells but also the subcellular distribution- Typically following phagocytosis, the particles should show up in the perinuclear region (For example see: Kim J, Sinha S, Solomon M, et al. Co-coating of receptor-targeted drug nanocarriers with anti-phagocytic moieties enhances specific tissue uptake versus non-specific phagocytic clearance. Biomaterials. 2017;147:14-25.)

9.       The authors claim “Human embryonic kidney (HEK 293T) cells and human hepatocytes(LO2 cells) were used to predict the  liver and kidney toxicity of RBGPs in vivo”. However, a cytotoxicity test against 2 cell lines does not predict what will happen in vivo with blood flow, organ distributions, metabolism and excretion processes, all occurring simultaneously. Furthermore, this method needs to be described in more detail- How was the survival rate calculated? Were there negative and positive controls for the cytotoxicity assay?

10.   For the hemolysis test- What is the read-out of the assay- Is it absorbance or some other spectrophotometric measurement? Visible color of hemolysin may not be an accurate measurement of release since small amounts may not be visible to the naked eye but may still be caused by the therapeutic.  Are the concentrations of berberine used for this test at clinically approved doses of the drug?

11.   The authors state “All these results implied that the release of BH was slow down in the BGPs group while the RBGPs could perform better. The release of BH from BGPs would be slow down further by physical barrier around the BGPs formed by RBCM.” These statements seem contradictory- Does perform better mean slower release or faster?

12.   In conclusions, the authors state that “…and increase the biocompatibility of BH.” Does it really increase the biocompatibility or the circulating time and the longer it circulates, the longer it can sustain its effect. Also the authors claim that this formulation can help overcome the low bioavailability of berberine- Is this true? Bioavailability is the fraction of the dose administered that is absorbed (makes it to the blood stream). As such, this is not a valid term for intravenously injected formulations which have 100% bioavailability. Unless another route of administration is to be compared between the berberine alone and this formulation, this conclusion does not apply.

Minor comments:

1.       Please have the manuscript edited by a native English speaker. There are quite a few instances where the sentence structure does not convey any meaning or is confusing. For instance use …Erythrocyte membrane-encapsulated celecoxib… instead of “celecoxib-encapsulated erythrocyte membranes (CB-RBCMs) were reported to have a sustain release of CB”. Other such examples: “In Fig.6, Dil (red) and FITC (green) were corresponded to a different particle compartment, and overlapped in the same locations (around the blue nucleus)”, “As a drug carrier, gelatin has an obvious sustained release effect and certain tissue targeting characteristics, which can increase the therapeutic while reduce the side effects” and several other instances in the introduction section.

2.       Typos throughout the text: Some instances p5 line134- solicited, p9 line241- DLC, line116- ultrasonic, line 177- 105 cells? and several others

3.       Use of scientific language: Line343- maintained instead of “kept”, … It can rapidly distribute to organs… instead of “it can quickly enter organs and tissues” and other instances

4.       In method 2.6, what do the W1,2,3 refer to: Amounts = mass/volume/concentration- Clarify.

5.       In introduction- include a section about other camouflaging techniques in literature and how this is better.

Author Response

Dear Reviewer,

Thank you very much for your letter and the comments from the referees about our paper submitted to pharmaceutics (Manuscript ID: pharmaceutics-408160)! We have checked the manuscript and revised it according to the comments. We submit here the revised manuscript as well as a list of changes.

If you have any question about this paper, please don’t hesitate to let me know.

Sincerely yours,

Mingfeng Qiu

Major comments:

Point 1: The authors claim that an obvious core-shell structure was observed by TEM of the particles. However, from the current images this is not visible. Furthermore, the size of the particles seems smaller than 200 nm in the TEM images, barely spherical and very polydisperse. If there are images that better reflect the population, please use those.

Response 1: The size was smaller than that measured by DLS,this was mainly due to the process involved in the preparation of sample. In the case of the TEM methods, TEM images depicted the actual size at the dried state of sample, however the size measured by DLS was a hydrodynamic diameter, the nanoparticles showed a larger hydrodynamic volume due to solvent effect in the hydrated state. Furthermore, TEM images seemed very polydisperse, the reason may be the concentration was a little high, so multiple layers of nanoparticles were present in the field of view.

Point 2: In the preparation of Cy5 labeled gelatin, how is the unbound Cy5 removed?

Response 2:PBS was used to wash the unbound Cy5, and was repeated for 3 times.

Point 3:     In Figure 2, what is the X- axis? It might be a better representation to show averages and sems of size and PDI in a table. Secondly, is there a statistically significant increase in size on coating membranes? Otherwise the conclusions that size increases on coating with the RBC membranes by 16 nm does not hold true- It might just be a variability of the measurement.

Response 3: The X-axis showed the times that the experiments have been repeated. In the view of TEMmost nanoparticles coated membranes can been found which  could confirmed the data.

Point4:  In the stability testing, the authors test up to day 7 for the RBGPs- What happens after day 7?

Response 4: After day 7, the size and PDI start to increase obviously, however stability testing for 7 days is enough.

Point 5: Method 2.8 is not clear. The SDS PAGE protocol has not been described nor the method of detection of the protein bands. Since this is a protein-denaturing protocol, the only information it yields is about the relative content of the proteins in the preparations, not about the composition or the distribution, maintenance of structure etc. Hence we cannot predict based on this that the biological characteristics and function of RBC-membranes is maintained in the nanoparticles, as the authors claim in several instances.

Response 5: In fact, the SDS PAGE just improved the membrane proteins were preserved, and the biological characteristics and function of RBCM could be validated by Immune escape experiment. The error expressions in Method 2.8 and Results and discussion 3.2 were modified.

Point 6:In the in vitro cellular uptake studies, why would you expect these particles to be taken up by hek cells- there are no targeting molecules or internalization cues on the surface of the particles. In fact the RBGPs would appear as RBCs to the cell and RBCs do not normally internalize into cells. Why would they bind to the surface of HEK cells? In fact, based on the fluorescence microcopy images, they do not appear to internalize (Fig 6). Are there controls of GPs alone, RBCMs alone as well to really understand what is going on? Furthermore, in the DAPI images, there is non-nucleus associated blue staining on the membranes of cells- Why is this so? Overall it is not clear how this experiment would fit in with the hypothesis.

Response 6: In fact we hope that the erythrocyte membrane-encapsulated nanoparticles can inherit some of the advantages of red blood cells, not means it appear as RBCs. Essentially it’s still a kind of nanoparticle, so the RBGPs could be internalize into cells. (e.g. Su, Jinghan Sun, Huiping Meng, Qingshuo, etc. Long Circulation Red-Blood-Cell-Mimetic Nanoparticles with Peptide-Enhanced Tumor Penetration for Simultaneously Inhibiting Growth and Lung Metastasis of Breast Cancer. Adv. Funct. Mater, 2016, 26, 1243–1252)

Point 7: All legends need more description. Figures and legend should be stand-alone data. However in this case, only a title is visible. Please include a brief description of the method (time of incubations, type of readout) statistics for the graphs etc. following the title.

Response 7: Thank you for your suggestion. Some descriptions needed to supplement, and we have refined in the manuscripts.

Point 8: In figure 7, the images of the fluorescent channel are very dim- It is not clear if there is any fluorescent signal. Are there some brighter or high magnification images of the same? It should tell not only about association of the particles with the cells but also the subcellular distribution- Typically following phagocytosis, the particles should show up in the perinuclear region.

Response 8: We improved the contrast and changed the images. And the purpose of experiment was to confirm that after coating by RBCM, nanoparticles could avoid the recognition by macrophage cells. The subcellular distribution may be performed as an additional experiment.

Point 9: The authors claim “Human embryonic kidney (HEK 293T) cells and human hepatocytes(LO2 cells) were used to predict the  liver and kidney toxicity of RBGPs in vivo”. However, a cytotoxicity test against 2 cell lines does not predict what will happen in vivo with blood flow, organ distributions, metabolism and excretion processes, all occurring simultaneously. Furthermore, this method needs to be described in more detail- How was the survival rate calculated? Were there negative and positive controls for the cytotoxicity assay?

Response 9: The formula to calculate the survival rate has been completed in the manuscripts, and cytotoxicity test in vivo is trying to accomplish. The control group(CG) and the blank group(BG) were set for the cytotoxicity assay, and also literatures have also reported the materials we used are nontoxic.( e.g. MJ Chou, HY Yu, JC Hsia, etc. Highly Efficient Intracellular Protein Delivery by Cationic Polyethyleneimine-Modified Gelatin Nanoparticles. Materials 2018, 11(2), 301)

Point 10: For the hemolysis test- What is the read-out of the assay- Is it absorbance or some other spectrophotometric measurement? Visible color of hemolysin may not be an accurate measurement of release since small amounts may not be visible to the naked eye but may still be caused by the therapeutic.  Are the concentrations of berberine used for this test at clinically approved doses of the drug?

Response 10: The percentage of hemolysis was measured by ultraviolet spectrophotometer. Each group was less than 0.1%, and the effective dose was in the intermediate concentration range, the hemolysis rate was less than 0.05%. It could be considered as no hemolysis

Point 11: The authors state “All these results implied that the release of BH was slow down in the BGPs group while the RBGPs could perform better. The release of BH from BGPs would be slow down further by physical barrier around the BGPs formed by RBCM.” These statements seem contradictory- Does perform better mean slower release or faster?

Response 11: Thank you for your suggestion. The unclear parts of the paragraph have been revised.

Point 12: In conclusions, the authors state that “…and increase the biocompatibility of BH.” Does it really increase the biocompatibility or the circulating time and the longer it circulates, the longer it can sustain its effect. Also the authors claim that this formulation can help overcome the low bioavailability of berberine- Is this true? Bioavailability is the fraction of the dose administered that is absorbed (makes it to the blood stream). As such, this is not a valid term for intravenously injected formulations which have 100% bioavailability. Unless another route of administration is to be compared between the berberine alone and this formulation, this conclusion does not apply.

 Response 12: Whether the system has good biocompatibility needs to be further verified by in vivo experiments. The conclusion has been modified to be clearer. The efficacy and other experiments will be proceeding in subsequent experiments in the future.

Minor comments:

Point 1: Please have the manuscript edited by a native English speaker. There are quite a few instances where the sentence structure does not convey any meaning or is confusing. For instance use …Erythrocyte membrane-encapsulated celecoxib… instead of “celecoxib-encapsulated erythrocyte membranes (CB-RBCMs) were reported to have a sustain release of CB”. Other such examples: “In Fig.6, Dil (red) and FITC (green) were corresponded to a different particle compartment, and overlapped in the same locations (around the blue nucleus)”, “As a drug carrier, gelatin has an obvious sustained release effect and certain tissue targeting characteristics, which can increase the therapeutic while reduce the side effects” and several other instances in the introduction section.

Point 2: Typos throughout the text: Some instances p5 line134- solicited, p9 line241- DLC, line116- ultrasonic, line 177- 105 cells? and several others

Point 3: Use of scientific language: Line343- maintained instead of “kept”, … It can rapidly distribute to organs… instead of “it can quickly enter organs and tissues” and other instances

Response1-3: Thank you for your comments, and we’ve modified the manuscripts according to your suggestions. For a few sections, we’ve rewritten the paragraphs to express more clearly, and some details have been modified at the same time. After the revision, we expect to achieve a more perfect English expression. We’d like to express our heartfelt gratitude for your helpful comments.

Point 4: In method 2.6, what do the W1,2,3 refer to: Amounts = mass/volume/concentration- Clarify.

Response 4 : Amounts means mass, we have clarified it.

Point 5: In introduction- include a section about other camouflaging techniques in literature and how this is better.

Response 5: In the paper we prefer to feature the RBCM, so we briefly list some other camouflaging techniques and supplemented the advantages of erythrocyte membrane in introduction.

Reviewer 3 Report

In this manuscript, Jing Su and collegues describe how, using erythrocyte membrane-camouflaged berberine hydrochloride-loaded gelatin nanoparticles, they slow down the release rate, increase the biocompatibility and reduce the side effects of BH. They describe clearly and in detail the synthesis and the characterization of RBGPs. 

According to my humble opinion, the manuscript is suitable for publication on  Pharmaceutics after minor revision.

Par.2.2 Cell culture: All the growth media contain only FBS?No glutamine or antibiotics?

Lane 126: at 4°C is repeated, please delete one.

Lane 137: BGP stands for? I think that the authors never write the entire name into the text.

Paragraph 2.8 (lanes 160-165): please enrich this paragraph with details as %of used gel, method for proteins detection.

Lane 172: I think that the cells are incubated with the substances, not the other way around. The Authors should check.

Lanes 235-236: please check, there is a repetition

Lanes 248-249 and fig.5. In the text, the authors say  :"SDS-PAGE gel electrophoresis test of RBGPs, BGPs, and natural RBCM were 249 performed..." but in fig.5 the loaded samples are different (RGPs, natural RBCM and GPs. Please verify. Again, do the loaded samples have the same protein concentration? If yes, why the RGPs samples produce a signal stronger than signal of the  natural RBCM samples?

Lane 252: RGP stands for?

In Fig.7   it is impossible to see the red fluorescence. Please improve the contrast.

Other questions:

-have the authors analyzed if hemolysis and aggregation vary with the variation in concentration?

-Have they a optimal range of concentration to use?

Author Response

Dear Reviewer,

Thank you very much for your letter and the comments from the referees about our paper submitted to pharmaceutics (Manuscript ID: pharmaceutics-408160)! We have checked the manuscript and revised it according to the comments. We submit here the revised manuscript as well as a list of changes.

If you have any question about this paper, please don’t hesitate to let me know.

Sincerely yours,

Mingfeng Qiu

Point 1:Par.2.2 Cell culture: All the growth media contain only FBS? No glutamine or antibiotics?

Response 1: The growth medias contain FBS as well as DMEM high sugar medium which meet the nutritional needs of cells

Point 2: Lane 126: at 4°C is repeated, please delete one.

Point 6:Lanes 235-236: please check, there is a repetition

Response 2 and 6: Thank you for scrutinize, we checked the manuscripts and revised the details.

Point 3:Lane 137: BGP stands for? I think that the authors never write the entire name into the text.

Point 8:Lane 252: RGP stands for?

Response 3 and 8 : Thank you for scrutinize, we checked the manuscripts and added the entire name when those are first mentioned.

Point 4:Paragraph 2.8 (lanes 160-165): please enrich this paragraph with details as %of used gel, method for proteins detection.

Response 4 :We have rewritten the paragraph with details of the method.

Point 5:Lane 172: I think that the cells are incubated with the substances, not the other way around. The Authors should check.

Response 5 : The cells are incubated with the substances, only some fluorescence labeling was performed before incubation.

Point 7:Lanes 248-249 and fig.5. In the text, the authors say  :"SDS-PAGE gel electrophoresis test of RBGPs, BGPs, and natural RBCM were 249 performed..." but in fig.5 the loaded samples are different (RGPs, natural RBCM and GPs. Please verify. Again, do the loaded samples have the same protein concentration? If yes, why the RGPs samples produce a signal stronger than signal of the natural RBCM samples?

Response 7: Thank you for your suggestions, but the sentence you had mentioned could not been found in the manuscript. However some mistakes really existed and we have modified them. For the second question, the loaded samples had the same protein concentration. SDS-PAGE was a qualitative experiment, the difference of signals between RGPs and the natural RBCM may be due to a small error.

Point 9:In Fig.7   it is impossible to see the red fluorescence. Please improve the contrast.

Response 9 : We have improved the brightness and changed the images. Thank you for your suggestions.

Other questions:

Point1:-have the authors analyzed if hemolysis and aggregation vary with the variation in concentration?

Point 2:-Have they a optimal range of concentration to use?

Response : In the experiment concentration was varied. The drug concentration in the nanoparticles is fixed, so we added different volume to change the final concentration of the whole solution, which had been shown in table 1.

Round 2

Reviewer 2 Report

The authors of the manuscript have made some comments and changes to try and improve the language of the manuscript but I still do not see a significant improvement to the manuscript, especially some missing controls and proper method descriptions. Let me address these points further. I have maintained the same numbering as my earlier comments and reply from authors for easy tracking.

Point 2: The removal of unbound cy5 is not addressed in the section of preparation of Cy5 labeled gelatin (Section 2.3). Let me reiterate.. Please describe the method by which you are removing the excess cy5-NHS from the gelatin solution after conjugation.

Point 3: I still do not see any mention of statistics which can be as simple as a Student’s T-test. Statistically sound science is a necessity for sound conclusions.

Point 5: The method 2.8 still does not address the method of detection of the protein bands.. Simply stating that the gel was stained does not make it clear to the reader what stain you are using- Coomassie blue, silver stain, etc.? Also, what gel imager did you use? Furthermore, the authors have still left the result that we can predict based on these results that the function of the RBC-membranes is retained in the nanoparticles (lines 297-298, line 328-329). Once again this is not true- Only the lack of uptake by macrophages suggests this.

Point 6: The article suggested by the authors as an explanation for the uptake is actually verifying the point I was trying to make. In this cited paper the targeting peptide, iRGD, is what enhances uptake into the tumor site (not the blank nanoparticle itself). Actually the tumor penetration is not a feature of the nanoparticles by themselves but a feature of iRGD coadministration, which can also enhance penetration of small molecules. In the case of the nanoparticles in this paper, what is the targeting moiety? Per the methods I only see that gelatin NPs are made without any reference to any peptide or other targeting moiety on them or coadministered with them. Again, the authors have not addressed or provided the controls that would be necessary to justify this as stated in my earlier point 6.

Point 7: The legends are still not detailed enough. An example legend would be… “Fig.5. Protein detection in RGPs, GPs and natural RBCM by SDS-PAGE followed by [method] staining of [protein amount loaded] ug fractions loaded in a ---% polyacrylamide gel.” Please modify other legends to a similar level of detail.

Point 8: The fluorescent channel images are still not brighter and are not acceptable for a publication because we cannot conclude anything.

Point 9: Define in the method what the control group and blank group are.

Point 10: State the wavelength used in the measurement of hemolysis. Only stating the equipment used does not suffice for a publication, especially if someone wants to replicate your work.

Minor comments that have still not been addressed:

Point 1: While the authors have tried to address some of the instances I brought up, they have not made an effort to clean up the whole text similarly. There are several instances where the language is still not refined (some examples only): line 38: distribute to, line 79: sustained release, line 87: What does it mean sustain release slowly?, line 213: washed, line 258: incubating for, Fig 2: repeated three times, line 322: internalization, line 395: for 7 d,

Point 2: The right term to use is sonicated, not soniced (2 instances to be changed).. line 212: Do you mean 105 cells? Please pay attention to detail. line 321: confocalization?

Point 3: Please eliminate the use of the word “obviously” to describe results. It is a non-scientific word that does not convey the correct meaning. There are many instances (line 93 , 91,  283, 397). Line 391: In summary, a new ….slow down the release rate [of what]?

Author Response

Dear Reviewer,

Thank you very much for your letter and the comments from the referees about our paper submitted to pharmaceutics (Manuscript ID: pharmaceutics-408160). We have checked the manuscript and revised it according to the comments. We submit here the revised manuscript as well as a list of changes.

If you have any question about this manuscript revision, please don’t hesitate to let me know.

Sincerely yours,

Mingfeng Qiu

Detailed response to Referee’s comments

Point 2: The removal of unbound cy5 is not addressed in the section of preparation of Cy5 labeled gelatin (Section 2.3). Let me reiterate. Please describe the method by which you are removing the excess cy5-NHS from the gelatin solution after conjugation.

Response: Thank you for your comments. In order to remove the excess Cy5, the solution was washed with PBS by centrifuging (13200 rpm10 min4 ), then it was repeated 3 times. Please forgive our carelessness that we have mistaken it in Section 2.9. This time we have supplemented Section 2.3. Please check and find it.

Point 3: I still do not see any mention of statistics which can be as simple as a Student’s T-test. Statistically sound science is a necessity for sound conclusions. ResponseThank you for your helpful comments and we’ve supplemented statistical analysis according to your valuable suggestion. The legend of Fig. 2 and a new section have been supplemented. It can be seen in Section 2.13:

Data expression was shown as the mean±SD. Significant differences between NRBCs and BSP-RBCs were analyzed by the Tukey Kramer multiple comparison test, using GraphPad Prism Software, v.6.01 (GraphPad Software, Inc.). Results with p < 0.05 were considered signifcant difference and very signifcant difference with p < 0.01.

Point 5: The method 2.8 still does not address the method of detection of the protein bands. Simply stating that the gel was stained does not make it clear to the reader what stain you are using- Coomassie blue, silver stain, etc.? Also, what gel imager did you use? Furthermore, the authors have still left the result that we can predict based on these results that the function of the RBC-membranes is retained in the nanoparticles (lines 297-298, line 328-329). Once again this is not true- Only the lack of uptake by macrophages suggests this.

ResponseThank you for your valuable comments, we’ve detailed the operation in this study and modified the inaccurate expression according to your helpful suggestions. Section 2.8 is now read as follows:

     Prepare the separation gel with a concentration of 8% and the concentrated gel with a concentration of 5%. The protein concentration of each sample was determined by BCA protein quantification kit and protein concentrations of all sample. RBCM, GPs, and RBCM-coated GPs (RBCM-GPs, RGPs) were adjusted to same concentration. Then 100 μL of the above samples of different groups were mixed with 25 μL loading buffer respectively. The mixtures were boiled in water bath at 95 °C for 5 min to make the proteins denatured. Then the prepared gel plate was loaded with 15 μL of each group and 2 μL protein marker was added into another separate loading slot. After the electrophoresis, the gel pieces were stained with Coomassie Brilliant Blue at room temperature for 30 min in the dark and decolorized overnight on decolorizing shaker. The gel was recorded by photographing under Tanon2500 Gel Imaging Analysis System (Shanghai Tianneng Technology Co., Ltd.).

Section 3.2 is now read as follows:

As shown in Fig. 5, protein bands of the natural RBCM group were almost same as that of the RGPs group, which means the membrane proteins of RBCM were maintained after the sonication treatment. This finding suggests that the associated membrane proteins were also transferred to the nanoparticle surface when the translocation of the bilayer cellular membranes occurred. We can expect that biological characteristics and function of the RBCMs could also be maintained, which could help the NPs to escape immune system.

Point 6: The article suggested by the authors as an explanation for the uptake is actually verifying the point I was trying to make. In this cited paper the targeting peptide, iRGD, is what enhances uptake into the tumor site (not the blank nanoparticle itself). Actually the tumor penetration is not a feature of the nanoparticles by themselves but a feature of iRGD coadministration, which can also enhance penetration of small molecules. In the case of the nanoparticles in this paper, what is the targeting moiety? Per the methods I only see that gelatin NPs are made without any reference to any peptide or other targeting moiety on them or coadministered with them. Again, the authors have not addressed or provided the controls that would be necessary to justify this as stated in my earlier point 6.

ResponseWe’d like to thank the reviewer for the helpful comment very much. After delving into your suggestion, we’re considering that the experiment may indeed be less rigorous, so we’ve temporarily deleted this section and will refine it in the future.

Point 7: The legends are still not detailed enough. An example legend would be… “Fig.5. Protein detection in RGPs, GPs and natural RBCM by SDS-PAGE followed by [method] staining of [protein amount loaded] ug fractions loaded in a ---% polyacrylamide gel.” Please modify other legends to a similar level of detail.

ResponseThanks a lot for your valuable suggestion. The legends of Fig. 2, Fig. 4, Fig. 5, Fig. 6 and Fig. 7 were modified again with details. Please check and find them.

Point 8: The fluorescent channel images are still not brighter and are not acceptable for a publication because we cannot conclude anything.

ResponseThank you very much for your helpful comment. We adjust the brightness and contrast again, meanwhile we try to point out the fluorescence with white arrows and hope it could be helpful for identifying the fluorescence.

Point 9: Define in the method what the control group and blank group are

ResponseWe’d like to thank the reviewer for the helpful comments very much. The control group only added DMEM medium, cells and CCK-8) while the blank group only added DMEM medium and CCK-8. We’ve supplemented the definitions in the articleSection 2.10. Please check and find it.

Point 10: State the wavelength used in the measurement of hemolysis. Only stating the equipment used does not suffice for a publication, especially if someone wants to replicate your work.

ResponseThank you for your careful comment to the missing part. The detection wavelength was 540nm, it has been supplemented. Please check and find it.

Minor comments that have still not been addressed:

Point 1: While the authors have tried to address some of the instances I brought up, they have not made an effort to clean up the whole text similarly. There are several instances where the language is still not refined (some examples only): line 38: distribute to, line 79: sustained release, line 87: What does it mean sustain release slowly?, line 213: washed, line 258: incubating for, Fig 2: repeated three times, line 322: internalization, line 395: for 7 d,

Point 2: The right term to use is sonicated, not soniced (2 instances to be changed).. line 212: Do you mean 105 cells? Please pay attention to detail. line 321: confocalization?

Point 3: Please eliminate the use of the word “obviously” to describe results. It is a non-scientific word that does not convey the correct meaning. There are many instances (line 93 , 91,  283, 397). Line 391: In summary, a new ….slow down the release rate [of what]?

ResponseThank you very much for your valuable comments. There are many mistakes in details and we’ve modified the manuscripts according to your suggestions again. Furthermore, we have revised the manuscript as well. After the revision, we expect to achieve a more perfect English expression. We’d like to express our heartfelt gratitude for your helpful comments.

Round 3

Reviewer 2 Report

The authors have considerably improved the manuscript and it is now acceptable for publication. There is only one minor change/correction: Add the ** symbol to the bars in figure 2. The legend states this but I do not see it in the figure. And just to give a final read for English language. 

Author Response

Dear Reviewer,

Thank you very much for your letter and the comments from the referees about our paper submitted to pharmaceutics (Manuscript ID: pharmaceutics-408160). We have checked the manuscript and revised it according to the comments. We submit here the revised manuscript as well as a list of changes.

If you have any question about this manuscript revision, please don’t hesitate to let me know.

Sincerely yours,

Mingfeng Qiu

Detailed response to Referee’s comments

Point:The authors have considerably improved the manuscript and it is now acceptable for publication. There is only one minor change/correction: Add the ** symbol to the bars in figure 2. The legend states this but I do not see it in the figure. And just to give a final read for English language. 

ResponseThank you for your careful comment to the missing part. The ** symbol has been added to the bars in figure 2. Please check it.
